# High Permittivity Polymer Composites on the Basis of Long Single-Walled Carbon Nanotubes: The Role of the Nanotube Length

**DOI:** 10.3390/nano12193538

**Published:** 2022-10-10

**Authors:** Shamil Galyaltdinov, Ivan Lounev, Timur Khamidullin, Seyyed Alireza Hashemi, Albert Nasibulin, Ayrat M. Dimiev

**Affiliations:** 1Laboratory for Advanced Carbon Nanomaterials, Chemical Institute, Kazan Federal University, 18 Kremlyovskaya Street, 420008 Kazan, Russia; 2Institute of Physics, Kazan Federal University, 18 Kremlyovskaya Street, 420008 Kazan, Russia; 3Nanomaterials and Polymer Nanocomposites Laboratory, School of Engineering, University of British Columbia, Kelowna, BC V1V 1V7, Canada; 4Skolkovo Institute of Science and Technology, Nobel Str. 3, 143026 Moscow, Russia

**Keywords:** carbon nanotubes, dielectric polymer composites, permittivity

## Abstract

Controlling the permittivity of dielectric composites is critical for numerous applications dealing with matter/electromagnetic radiation interaction. In this study, we have prepared polymer composites, based on a silicone elastomer matrix and Tuball carbon nanotubes (CNT) via a simple preparation procedure. The as-prepared composites demonstrated record-high dielectric permittivity both in the low-frequency range (10^2^–10^7^ Hz) and in the X-band (8.2–12.4 GHz), significantly exceeding the literature data for such types of composite materials at similar CNT content. Thus, with the 2 wt% filler loading, the permittivity values reach 360 at 10^6^ Hz and >26 in the entire X-band. In similar literature, even the use of conductive polymer hosts and various highly conductive additives had not resulted in such high permittivity values. We attribute this phenomenon to specific structural features of the used Tuball nanotubes, namely their length and ability to form in the polymer matrix percolating network in the form of neuron-shaped clusters. The low cost and large production volumes of Tuball nanotubes, as well as the ease of the composite preparation procedure open the doors for production of cost-efficient, low weight and flexible composites with superior high permittivity.

## 1. Introduction

Controlling dielectric permittivity of polymer composites is critical for thin-film transistors [1], photovoltaic devices [2] and more broadly for materials aimed at absorption/reflection of electromagnetic radiation, especially in the X-band [3,4,5,6]. The most commonly used polymer matrices are silicon rubber [7,8,9,10,11,12,13,14,15], epoxy resin [16,17,18,19,20], polyvinylidene fluoride (PVDF) [21,22,23,24], and thermo-polyurethane (TPU) [25,26]. Correspondingly, the most broadly used conductive fillers are carbon nanotubes (CNTs), metal particles and graphene derivatives [4,5,6,7,8,9,10,11,12,13,14,15,16,17,18,19,20,21,22,23,24,25,26,27,28]. Among the fillers for this aim, CNTs are of special interest since they possess high aspect ratio, high electrical conductivity and superior mechanical strength [29]. In the literature, there are more papers on the use of multi-walled carbon nanotubes (MWCNTs) [6,7,8,9,10,13,14,15,16,17,21,22,23,24] rather than single-walled carbon nanotubes (SWCNT) [30,31,32,33,34]. This is because MWCNTs are more available, significantly cheaper and can be more easily and uniformly distributed in the polymer matrices.

Among all the SWCNTs commercially available on the market, nanotubes sold under the brand Tuball manufactured by OCSiAl are of particular interest [33,34,35]. This is because they are currently manufactured at a tonnage scale and are offered at the cheapest price. A structural feature of these nanotubes is their large diameter (1.4–2.2 nm), compared to other SWCNTs such as HiPCo, CoMoCat, etc., and very large length [36]. Such unique features make them attractive as fillers [37,38] and additives to polymers and concrete [39].

The two-component composites, in which a dielectric polymer host contains only CNTs as a conductive filler, exhibit relatively low dielectric permittivity values (*ε*′) even in the low-frequency range [7,9,11,12,13,14,15]. Higher dielectric permittivity values are normally achieved using either a conductive polymer host [22,23,24] or/and special additives [31]. The maximum permittivity value was achieved in [21] with PVDF matrix and modified CNTs; at the 3.5 wt% CNT content, the resulting composite exhibiting permittivity values of about 250 at 10^4^ Hz.

In the X-band, for the two-component systems made of epoxy resin and MWCNTs, in most studies, the reported permittivity values do not exceed 13 [18,26,40]. The higher value of *ε*′ > 20 was obtained when using a higher MWCNT content (5 wt%) [26]. To the best of our knowledge, the highest reported value of *ε*′ = 23 at 3% CNT content was attained when using MWCNTs and a polar dielectric matrix, such as PVDF [41]. Alternative ways to increase the dielectric permittivity include addition of a third highly conductive component such as noble metals [6,42], which increases the cost of the final material. At the same time, for the two-component composites, consisting of a non-polar dielectric matrix and SWCNTs as conductive filler, high dielectric permittivity values have not been reported yet.

In this study, for the first time, we report the dielectric properties of the silicone elastomer-based composites reinforced with Tuball SWCNTs. The composites demonstrate tremendously high dielectric permittivity values both in the low-frequency range and in the X-band. Importantly, such values were obtained using affordable components by a simple preparation method without high-cost additives.

## 2. Experimental

### 2.1. Materials

The SWCNTs were of the Tuball brand manufactured by OCSiAl (Luxembourg) (01RW03.N1, batch no. 819); SWCNTs were purified by the manufacturer. According to the manufacturer, the content of nanotubes was ≥93% and the content of metallic impurities was <1%. The ToolDecor T 20–137, a two-part molding silicone elastomer, was supplied from Wacker (Munchen, Germany); it consists of two components, ToolDecor T 20–1 Base (Part A) and Catalyst T 37 Hardener (Part B).

Methylene chloride was purchased from Tatkhimprodukt LLC (Kazan, Russia) and used without additional purification.

### 2.2. Characterization

Raman spectra of nanotubes were acquired from the nanotube films using an ARS-3000 Raman microscope (NanoScanTechnologies, Russia) with the 532 nm excitation laser. The scanning electron microscopy (SEM) images were acquired with a Merlin field-emission high resolution scanning electron microscope (Carl Zeiss, Oberkochen, Germany) at accelerating voltage of incident electrons of 5 kV and current probe of 300 pA.

### 2.3. Preparation of Polymer Composites

To prepare reinforced composites, nanotubes in precalculated quantities were dispersed in CH_2_Cl_2_ by sonication with a tip sonicator Sonic-Vibra 750 (Sonics, Newtown, CT, USA) for 1 h at 30% amplitude. Methylene chloride was chosen as a solvent, because it can dissolve silicone and has a low boiling point, facilitating its subsequent evaporation. Then the resulting dispersion of nanotubes was mixed with part A of the silicone elastomer and agitated manually with a glass rod until homogeneous condition. Next, the mixture was mildly sonicated for 30 min at 20% amplitude. After that, the as-obtained paste was heated in a water bath at T = 60–70 °C with manual agitation until complete evaporation of CH_2_Cl_2_. The resulting paste was thoroughly mixed with part B in a weight ratio of 100A:4B and placed into silicone molds. The samples were cured for 12 h at room temperature. As a result of these operations, samples of composites in the form of disks were obtained: for the low-frequency measurements, the disks were fabricated with a diameter of 29.0 mm and a thickness of 3.5 mm; for the high-frequency measurements, the disks were fabricated with a diameter of 29.0 mm and a thickness of 7 mm.

### 2.4. Electrical Measurements

The permittivity and loss values for the low-frequency range were calculated from the capacitance, measured with the Novocontrol BDS Concept-80 impedance analyzer, (Novocontrol Technologies GmbH & Co. KG, Montabaur, Germany) with the automatic temperature control provided by the QUATRO cryo-system (the temperature uncertainty is ±0.5 °C). A sample was placed between two gold-plated electrodes of the capacitor. The capacitor was attached to the thermostated testing head. The measurements were conducted in the frequency range of 0.1 Hz–10 MHz. The data for the ultra-high frequency (UHF) range (0.1–70 GHz) were measured with the PNA-X Network Analyzer N5247A (Agilent Technologies, Santa Clara, CA, USA). Samples in the form of disks with a diameter of 29 mm and a thickness of 7 mm were placed at the end of a coaxial measuring probe (Performance Probe) with a diameter of 10 mm. When measuring on the PNA-X Agilent N5247A, the results were recorded using the Agilent 85070 built-in licensed software package (Santa Clara, CA, USA). The temperature was set at 25 °C. The processing of the experimental data was carried out with the WinFit software [43].

## 3. Results and Discussion

The used Tuball CNTs were characterized by SEM and Raman spectroscopy (Figure 1). The SEM images (Figure 1a) reveal that nanotubes exist mainly in the form of thick bundles consisting of tens and hundreds of individual nanotubes. According to the higher magnification SEM and TEM images [38], the primary bundles with diameters of 5–10 nm are assembled into secondary and even tertiary bundles with diameters of up to 50 nm. The existence of Tuball nanotubes in the form of bundles is a consequence of their longer length compared to other commercially available SWCNTs, such as HiPCo, CoMoCat, etc. [36].

The Raman spectrum is typical for SWCNTs. In addition to the two-component G-band, there is a small D-band in the vicinity of ~1336 cm^−1^ (Figure 1b), which shows the presence of defects in the crystal lattice. The D-peak intensity is slightly higher than that in the as-received SWCNTs. Some additional defects were introduced during the sonication of the CNT dispersions. Radial breathing mode (RBM) is sensitive to the nanotube diameter [44,45] and coincides with the literature data for Tuball CNTs. The provided characteristics (Figure 1) are sufficient for the purposes of this study. For a more detailed description of the used CNTs, we refer to our previous publication [35].

Figure 2 shows the real (*ε*′) and imaginary (*ε*″) parts of the complex permittivity of the composites with different filling fractions as a function of frequency. Real permittivity values in the range of 0.1–300 Hz are not shown in Figure 2, and are not considered in the approximation of the curves since the material in this frequency range is very sensitive to mechanical deformations that arise upon its contact with the electrodes of the measuring capacitor. The full range dielectric spectra are shown in the Appendix A.

In general, an increase in the CNT content leads to an increase in the dielectric permittivity in the whole tested frequency range. The composites can be divided into three groups according to the proximity of the permittivity values:
(A)Group I—composites with a filler content of 0.1–0.5%;(B)Group II—composites with a filler content of 1 and 1.5%;(C)Group III—composites with a filler content of 2 and 3%.

For the composite with a filler content of 0.1%, *ε*′ practically does not change over the entire frequency range. For the composites with a filler content of 0.25% and 0.5%, a smooth increase in *ε*′ in the range of 1 × 10^7^–3 × 10^2^ Hz is observed without reaching a plateau at the low frequency end. For the Group II composites with a filler content of 1–1.5%, the growth of *ε*′ values is observed in the range of 1 × 10^5^–1 × 10^7^ Hz, followed by a plateau below 5 × 10^4^ Hz with *ε*′ values around 280. Finally, for the Group III composites, a smooth increase in the *ε*′ values is observed in the range of 3 × 10^5^–1 × 10^7^ Hz with a plateau starting at 2.5 × 10^5^ Hz with *ε*′ values reaching 470 at 1 × 10^4^ Hz.

The imaginary parts of the complex permittivity (Figure 2b) specify the dielectric losses. In the tested frequency range, no loss peak is registered for the composite with a filler content of 0.1%. For the 0.25% sample, a broad peak in the range of 3 × 10^6^–1 × 10^6^ Hz is observed. For the composite with a 0.5% CNT content, a broad peak is centered at 1.4 × 10^5^ Hz. The increase in the filler content leads to the shift of the loss peak position to the higher frequency region. Thus, for the 1% content, the loss peak is at 1.6 × 10^6^ Hz, for 2% content, the peak is at 6 × 10^6^ Hz, and for 3% content, the peak is located above 1 × 10^7^ Hz. In the curves of the composites with 3% CNT content, only the left shoulder of the loss peak is visible in the entire frequency range. The peak of the dielectric losses is originated by the polarization delay with an increase in the frequency of external electric field, and is normally indicative of so-called dipolar relaxation [46].

Table 1 compares the published literature data with the values of the dielectric permittivity obtained in this study [14,15,21,22,23,24,31]. According to the presented data, the composites prepared in this work significantly outperform the known literature data for the polymer/carbon-nanotubes systems with the same CNT content. Importantly, the attained permittivity values are much higher than the values reported in studies [14,15] on similar systems made from silicon elastomer matrix and multi-wall carbon nanotubes. Even the use of conductive matrices such as PVDF [21,22,23,24], the modification of nanotubes [21], and addition of polyaniline (PANI) [31] do not afford such high values of *ε*′, as we attain in this study.

Normally, permittivity increases with increasing the conductive filler fraction. Subsequently, at very high CNT content (>7%), expectedly higher permittivity values have been reported in literature. However, for the CNT content range, tested in this wok (1–3%), we are reporting the record high values (Table 1).

Such a high permittivity values must be related to the special structural features of these CNTs. Namely, due to their length, Tuball CNTs form bundles, and do not easily unbundle even with prolonged high power sonication [35,36]. After unbundling they tend to quickly rebundle, if are not stabilized by surfactants or by other means. Respectively, in the composites, these CNTs would exist in the form of bundles. In addition to bundles, in polymers, CNTs form aggregates, consisting of many bundles. To investigate the structure of our composites, we used optical microscopy. Figure 3 represents optical microphotographs of the liquid CNT/silicon elastomer formulations with different CNT contents taken before the formulations have been cured. It is clear that, even at the 0.25% CNT content, the CNT bundles are joined into clusters, which are well visible in the fully transparent silicon elastomer. The shape of the clusters resemble neurons with the tails of the CNT bundles sticking out from the main cluster body. With increasing the CNT content, not only the number, but also the size of the clusters increase. The size of the clusters vary from 20 µm through 200 µm.

The sample regions containing isolated clusters of nanotubes form conductive inclusions, randomly distributed in the polymer matrix. According to the knowledge in the field, the pertinent physics behind the polarization mechanism in composite materials remains poorly understood [46]. Considering the structure of the composite, namely the size of the CNT clusters and their distribution, as well as the presence of the loss peaks on the imaginary part functions (Figure 2b), the registered phenomenon might be least contradictorily explained in the terms of the Maxwell-Wagner polarization [20,22,24,33,46,47]. The larger the size of the clusters and their number, the larger the interface with the polymer matrix. Respectively, the higher charge can be accumulated at the interface. Such polarization in the larger clusters will fully manifest at lower frequencies.

To extract additional relaxation parameters, the measured dielectric spectra were approximated by the superposition of the Cole-Cole function [48] and the Jonsher parameter [46] according to the following equation:(1)$$\varepsilon^{*}\left(\omega \right)=\varepsilon^{\prime }\left(\omega \right)-i\varepsilon^{\prime \prime }\left(\omega \right)=\varepsilon_{\infty }+\frac{\Delta \varepsilon }{1+{\left(i\omega \tau \right)}^{\alpha }}+B{\left(i\omega \right)}^{n-1}+i\frac{\sigma_{0}}{\varepsilon_{0}\omega }$$
where *ε*′(*ω*) and *ε*″(*ω*) are the real and the imaginary parts of the complex permittivity; *i* is the imaginary unit; Δ*ε* = *ε*_s_ − *ε*_∞_, *ε*_s_ is the static dielectric permittivity; *ε*_∞_ is the permittivity at high frequency; *ω* is the cyclic frequency, Δ*ε, τ, α* are the magnitudes of the dielectric strength, relaxation time, and Cole-Cole broadening parameter, respectively; *B* is the magnitude of the Jonscher correction; 0 < *n* ≤ 1 is the Jonscher parameter; *σ*_0_ is the DC-conductivity; *ε*_0_ = 8.85 × 10^−12^ F/m is the permittivity of vacuum. The approximation of the dielectric spectrum by Equation (1) for a sample with a filler content of 1% is shown in Appendix A.

Figure 4a shows that the DC-conductivity *σ*_0_ smoothly increases with increasing the filler content in the composites that is in accordance with the literature values [7]. The dependence of the relaxation times on the filler concentration in the samples is shown in Figure 4b. In general, relaxation time decreases with an increase in the filler concentration.

Figure 5 shows the frequency dependence of dielectric permittivity of the same materials in the X-band region. The original dielectric spectra of the composites in the whole UHF region are presented in the Appendix A. There is no significant change in the dielectric permittivity with frequency within the 8–12 GHz range (Figure 5a). Expectedly, composites with a higher filler content have higher *ε*′ values. Based on the attained permittivity, the tested materials can be again divided into three groups with close *ε*′ values:
Icomposites with filler content of 0.1–1%;IIcomposite with filler content of 1.5%;IIIcomposites with filler content of 2–3%.

Comparison of our data with the published literature data in the X-band is presented in Table 2. Again, the permittivity values registered in this work notably exceed the literature data [18,26,40,41]. Even the use of the polar polymer host, such as PVDF [41], does not enable the values, attained in our study. At the same time, the values of the imaginary part of the dielectric permittivity, registered in this study (Figure 5b), are among the lowest in the literature [49].

Figure 6a presents an approximation of the real and imaginary parts of the complex permittivity for a sample with a nanotube content of 2%. The spectra were approximated by the Cole-Cole function with the Jonsher parameter as was shown above.

In contrast to the low-frequency spectra (Figure 2 and Figure 4), no signs of DC-conductivity are observed in the UHF region, since the processes characteristic of DC-conductivity do not have time to occur during this short time intervals. Apparently, the periods of the field oscillations are rather short, and relaxation processes might reflect the polarization occurring inside the conducting CNT bundle. We hypothesize that the size of the CNT bundles of the Tuball nanotubes is the main factor, responsible for the record high permittivity values in the X-band.

Figure 6b shows the dependence of the static dielectric permittivity (*ε_s_*) on the filler content in the UHF region. The *ε_s_* values were obtained by approximating the dielectric spectra using the Cole-Cole equation [46,48]. In the range of 0.1–1%, the dielectric permittivity changes insignificantly with the filler content. After 1%, a sharp increase in *ε_s_* is observed, followed by a plateau after >2%.

Figure 6c represents the function of the relaxation time τ on the filler content in the UHF region. The values of τ insignificantly change with increasing the filler content; that they have the same order of magnitude as opposed to the situation in the low-frequency region in which they differ ~10^5^ times in the tested range of the filler contents (Figure 4b).

Apparently, there must be a difference in the polarization mechanisms in the two tested frequency regions. At low frequencies, high permittivity can be explained by the Maxwell-Wagner polarization with possible contribution of the charge transfer by hopping electrons through thin polymer layers. However, at high frequencies, we most likely face another polarization mechanism.

## 4. Conclusions

In this study, polymer composites have been prepared by incorporating the Tuball single-walled carbon nanotubes into a silicone matrix via a simple cost-efficient preparation procedure. The registered dielectric permittivity values in the low-frequency range (3 × 10^2^−10^7^ Hz) and in the X-band significantly exceed the previously published literature data for the similar systems at similar CNT content. Even the use of polar polymer hosts and various highly conductive additives in the literature had not resulted in such high dielectric permittivity values. The high permittivity values registered in this study are the consequence of the specific features of the Tuball nanotubes, namely their high length and ability to join into long bundles, forming percolative clusters. The polarization mechanism in the two tested frequency ranges is suggested to be different.

## Figures and Tables

**Figure 1 nanomaterials-12-03538-f001:**
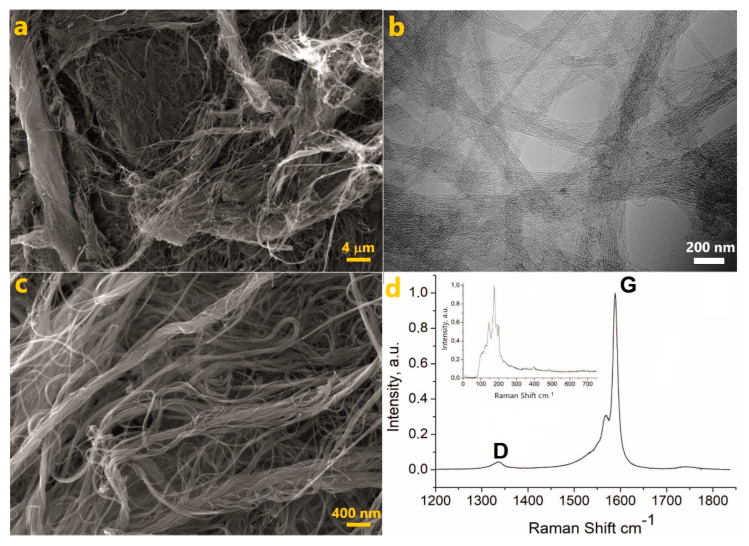
Characterization of as-received Tuball SWCNTs. (**a**,**c**) SEM images at different magnification. (**b**) TEM image. (**d**) Raman spectrum of CNTs. Tangential mode. The inset is the radial breathing mode. The spectra were acquired with the 532 nm excitation laser.

**Figure 2 nanomaterials-12-03538-f002:**
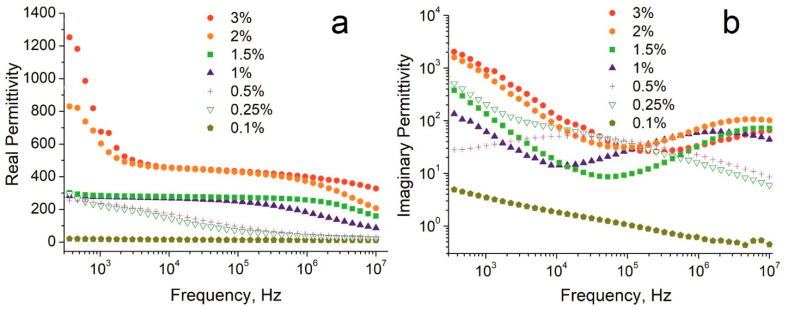
Frequency dependences of the (**a**) real and (**b**) imaginary parts of the complex permittivity for composites with different filler contents.

**Figure 3 nanomaterials-12-03538-f003:**
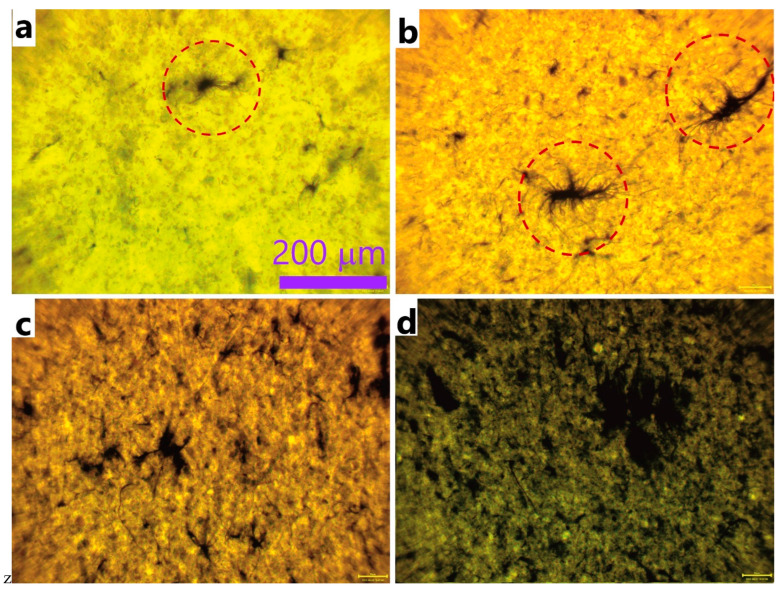
Optical microphotographs of liquid CNT/silicon elastomer formulations with different CNT content: 0.25% (**a**), 0.5% (**b**), 1.0% (**c**), and 2.0% (**d**). The scale bar is the same for all the four images. The red-line circles on (**a**,**b**) show the largest CNT clusters, present in the sample.

**Figure 4 nanomaterials-12-03538-f004:**
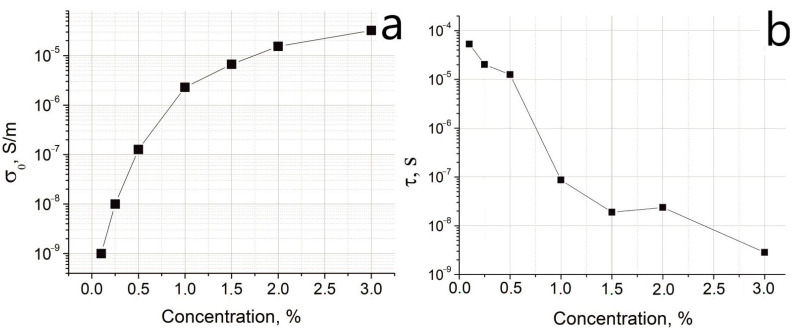
The approximation derived dependence of (**a**) DC-conductivity *σ*_0_, and (**b**) relaxation times *τ* on the content of CNTs for a sample with a filler content of 1% in the frequency range of 300–10^7^ Hz.

**Figure 5 nanomaterials-12-03538-f005:**
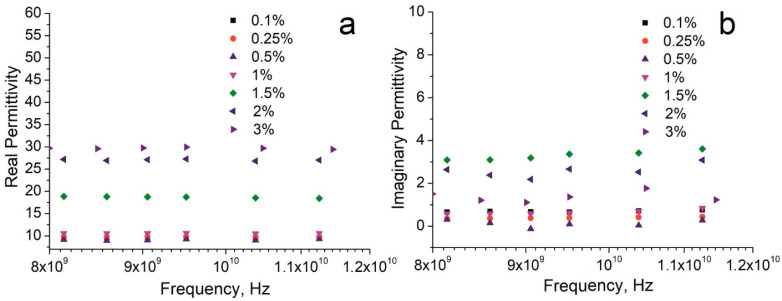
Frequency dependence of the (**a**) real and (**b**) imaginary parts of the complex permittivity for different filler contents in the X-band.

**Figure 6 nanomaterials-12-03538-f006:**
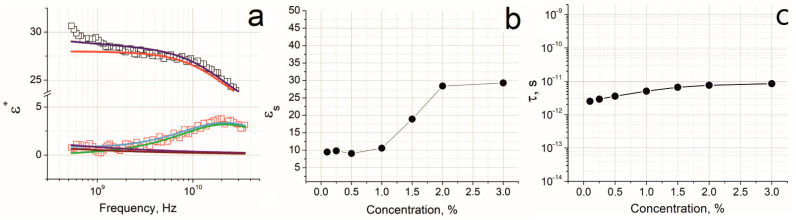
Approximation of the dielectric spectra for a sample with a nanotube content of 2% in the UHF region. (**a**) The empty black squares are the real part, and the empty red squares are the imaginary part of the complex permittivity. Blue and light blue lines are fitting functions for the real and imaginary parts of the spectrum, respectively; red and green lines are Cole-Cole functions for the real and imaginary parts of the spectrum, respectively; violet and brown lines are Jonsher corrections for the real and imaginary parts of the spectrum, respectively. (**b**) Dependence of the static dielectric permittivity on the filler content. (**c**) The function of the relaxation time on the filler content in the UHF region.

**Table 1 nanomaterials-12-03538-t001:** The literature data on the permittivity of different polymer composites, comprising CNTs as a conductive filler, in the frequency range of 10^4^–10^6^ Hz.

Polymer Host/Filler	CNT (wt%)	*ε*′ at 10^4^ Hz	*ε*′ at 10^6^ Hz	Ref.
PDMS/MWCNTs	3	~4.5	~4.3	[14]
Silicon rubber/MWCNTs	2.5	~4.5	~4.4	[15]
PVDF/functionalized MWCNTs	3.5	~250	~200	[21]
PVDF/MWCNTs	3.7	160	~100	[22]
PVDF/MWCNTs	2	~225	~50	[23]
PVDF/MWCNTs	4	~30	~25	[24]
TPU-PANI/SWCNTs	0.5	~100	~80	[31]
Silicon rubber/SWCNTs	2	~450	~360	This work

**Table 2 nanomaterials-12-03538-t002:** The literature data for the permittivity (*ε*′) of different composites made from nanotubes and polymer matrix in the X-band.

Polymer Host/Filler	CNT Loading Fraction, %	*ε*′	Ref.
Epoxy resin/MWCNTs	1	~4.75	[18]
TPU/MWCNTs	3	~13	[26]
Epoxy resin/MWCNTs	2	3.0	[40]
PVDF/MWCNTs	3	~23	[41]
Silicone/SWCNTs	2	~27	This work
Silicone/SWCNTs	3	~30	This work

## Data Availability

The data presented in this study are available on request from the corresponding author.

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
