# Peer review of "High Permittivity Polymer Composites on the Basis of Long Single-Walled Carbon Nanotubes: The Role of the Nanotube Length"

_nanomaterials, 2022, doi:10.3390/nano12193538_

Round 1

Reviewer 1 Report

Page 1 “A structural feature of these nanotubes is their large diameter (1.4–2.2 nm)…”  This would not be considered large diameters for nanotubes.

Delete Part in the label for section 2

Page 5 “It is clear that even at the 0.25% 194 CNT content, the CNT bundles are joined into clusters.”  I am not sure what features in Figure 3 suggest this so the features of the images need to be explained more clearly and perhaps the important features can be highlighted in the images.  Raman mapping which likely be a much better approach to this experiment—it would highlight the position of the CNTs.

Page 8 “Apparently, there must be a difference in the polarization mechanisms in the two 283 tested frequency regions. At low frequencies, high permittivity can be explained by the Maxwell-Wagner polarization with possible contribution of the charge transfer by hopping electrons through thin polymer layers. However, at high frequencies, we most likely face another polarization mechanism.”  More explanation of this comment would be helpful—why is this the case?

Author Response

Point by Point Response to Reviewers' Questions

We have addressed the questions, raised by reviewers, as fully as it was possible, considering the short timeframe given to us by the editor. In particular, we modified two Figures, and added/modified some text fragments in the manuscript.

Below are our point-by-point responses to the reviewer's questions. The reviewer's questions are given in Bold Italics. Our responds are given in normal fonts. The original manuscript text fragments are given in blue Italics. The newly added and corrected text is given in red Italics.

Reviewer 1

Page 1 “A structural feature of these nanotubes is their large diameter (1.4–2.2 nm)…”  This would not be considered large diameters for nanotubes.

Thank you for the comment. Sorry for not making ourselves clear. This diameter is large for SWCNTs. The text was revised as the following.

A structural feature of these nanotubes is their large diameter (1.4–2.2 nm), compared to other SWCNTs such as HiPCo, CoMoCat, etc., and very large length [36].

Delete Part in the label for section 2

Thank you. Deleted.

Page 5 “It is clear that even at the 0.25% 194 CNT content, the CNT bundles are joined into clusters.”  I am not sure what features in Figure 3 suggest this so the features of the images need to be explained more clearly and perhaps the important features can be highlighted in the images.  Raman mapping which likely be a much better approach to this experiment—it would highlight the position of the CNTs.

Per Reviewer’s request, we highlighted the CNT clusters on the optical images, and added more discussion on this matter in the text on page 9 as the following.

Figure 3 represents optical microphotographs of the liquid CNT/silicon elastomer formulations with different CNT contents taken before the formulations have been cured. It is clear that even at the 0.25% CNT content, the CNT bundles are joined into clusters, which are well visible in the fully transparent silicon elastomer. The shape of the clusters resemble neurons with the tails of the CNT bundles sticking out from the main cluster body. With increasing the CNT content, not only the number, but also the size of the clusters increase.  The size of the clusters vary from 20 µm through 200 µm.

The captions to Figure 3. Optical microphotographs of liquid CNT/silicon elastomer formulations with different CNT content: 0.25% (a), 0.5% (b), 1.0% (c), and 2.0% (d). The scale bar is the same for all the four images. The red-line circles on (a) and (b) show the largest CNT clusters, present in the sample.

 At the same time, we cannot perform Raman mapping in a short timeframe, provided to us by the Editor. Also, we do not think that the Raman mapping is critical here. Raman is a powerful informative tool, used to analyze structural details of CNTs. It is not really needed for the purpose to confirm the presence of the nanotubes. The provided optical images show the distribution of the CNTs and the shape of the CNT clusters with high clarity.

Page 8 “Apparently, there must be a difference in the polarization mechanisms in the two 283 tested frequency regions. At low frequencies, high permittivity can be explained by the Maxwell-Wagner polarization with possible contribution of the charge transfer by hopping electrons through thin polymer layers. However, at high frequencies, we most likely face another polarization mechanism.”  More explanation of this comment would be helpful—why is this the case?

Thank you for this comment. As admitted by the experts in the field, the pertinent physics behind the polarization of such composites remain largely unknown. This is why, in this manuscript, we purposefully avoided discussing the polarization mechanism, and only cautiously mentioned it in the cases where we are more or less confident. We cannot confidently do more than we did. Besides, discussing the polarization mechanism is the subject of physics journals. This special issue is aiming practical applications of CNTs, which we provided.

Reviewer 2 Report

In this work, the authors prepared polymer composites, based on a silicone elastomer matrix and Tuball carbon nanotubes (CNT) via a simple preparation procedure. the structure and performance are studied. this work can be accepted for publication after major revison.

1. The purity of reagents and raw materials is important for this work.

2. It is better to add the model of instruments and equipments.

3. The SEM image of CNTs at high magnification and TEM images of CNTs are important for the clearly observation.

4. The authors should provide the XRD pattern and add the corresponding discussion and analysis.

5. More structure characterizations and performance analysis can be added in the revised version.

Author Response

Point by Point Response to Reviewers' Questions

We have addressed the questions, raised by reviewers, as fully as it was possible, considering the short timeframe given to us by the editor. In particular, we modified two Figures, and added/modified some text fragments in the manuscript.

Below are our point-by-point responses to the reviewer's questions. The reviewer's questions are given in Bold Italics. Our responds are given in normal fonts. The original manuscript text fragments are given in blue Italics. The newly added and corrected text is given in red Italics.

Reviewer 2

In this work, the authors prepared polymer composites, based on a silicone elastomer matrix and Tuball carbon nanotubes (CNT) via a simple preparation procedure. the structure and performance are studied. this work can be accepted for publication after major revison.

  1. The purity of reagents and raw materials is important for this work.

The purity of CNTs might indeed affect the recorded permittivity values. It is fully provided in the Experimental part page 3 of the manuscript as the following.

The SWCNTs were the brand Tuball manufactured by OCSiAl (01RW03.N1, batch no. 819); SWCNTs were purified by the manufacturer. According to the manufacturer, the content of nanotubes was ≥93%, and the content of metallic impurities was <1%.

  1. It is better to add the model of instruments and equipments.

All the instrumentation, used in the work is fully described in the experimental part (section 2.4, page 4-5) as the following.

The permittivity and loss values for the low-frequency range were calculated from the capacitance, measured with the Novocontrol BDS Concept-80 impedance analyzer, with the automatic temperature control provided by the QUATRO cryo-system (the temperature uncertainty is ± 0.5 ° C). A sample was placed between two gold-plated electrodes of the capacitor. The capacitor was attached to the thermostated testing head. The measurements were conducted in the frequency range of 0.1 Hz – 10 MHz. The data for the ultra-high frequency (UHF) range (0.1 – 70 GHz) were measured with the PNA-X Network Analyzer N5247A (Agilent Technologies, USA). Samples in the form of disks with a diameter of 29 mm and a thickness of 7 mm were placed at the end of a coaxial measuring probe (Performance Probe) with a diameter of 10 mm. When measuring on the PNA-X Agilent N5247A, the results were recorded using the Agilent 85070 built-in licensed software package. The temperature was set at 25°C. The processing of the experimental data was done with the WinFit software [43].

  1. The SEM image of CNTs at high magnification and TEM images of CNTs are important for the clearly observation.

Per Reviewer’s recommendation we have added two new SEM images at different magnifications and a TEM image of the used CNTs to Figure 1.

  1. The authors should provide the XRD pattern and add the corresponding discussion and analysis.

Unfortunately, Reviewer did not specify why “authors should provide the XRD pattern”. What is the purpose? What characteristics of the samples Reviewer suggests to obtain? In our opinion, XRD is not applicable for our samples and for our study, since our samples do not yield any XRD pattern.

  1. More structure characterizations and performance analysis can be added in the revised version.

Unfortunately, Reviewer is not specific. We do not understand what exactly we are missing and what Reviewer wants us to demonstrate/discuss. In such circumstances, it is difficult to add something specific. We think that the provided characterization and discussion are sufficient to describe both made observations and conclusions.

Round 2

Reviewer 2 Report

This revised version can be accepted for publication.